# Decreased Hippocampal Neurogenesis in Aged Male Wistar Rats Is Not Associated with Memory Acquisition in a Water Maze

**DOI:** 10.3390/ijms241713276

**Published:** 2023-08-26

**Authors:** Mikhail Stepanichev, Victor Aniol, Natalia Lazareva, Natalia Gulyaeva

**Affiliations:** Institute of Higher Nervous Activity and Neurophysiology, Russian Academy of Sciences, Butlerova Str., 5a, Moscow 117485, Russia; aniviktor@yandex.ru (V.A.); nalaza@rambler.ru (N.L.); nata_gul@ihna.ru (N.G.)

**Keywords:** aging, neurogenesis, hippocampus, learning, memory, nitric oxide, neuronal NO-synthase

## Abstract

Brain aging is associated with a progressive decrease in learning abilities, memory, attention, decision making, and sensory perception. Age-related cognitive disturbances may be related to a decrease in the functional capacities of the hippocampus. This brain region is essential for learning and memory, and the lifelong neurogenesis occurring in the subgranular zone of the dentate gyrus may be a key event mediating the mnemonic functions of the hippocampus. In the present study, we investigated whether age-related changes in hippocampal neurogenesis are associated with learning and memory disturbances. Four- and 24-month-old rats were trained to find a hidden platform in a water maze. Though the older group showed higher latency to search the platform as compared to the younger group, both groups learned the task. However, the density of proliferating (PCNA-positive), differentiating (Dcx-positive), and new neurons (pre-labeled BrdU-positive) was significantly lower in the hippocampus of aged rats as compared to young ones. This inhibition of neurogenesis could be related to increased local production of nitric oxide since the density of neurons expressing neuronal NO-synthase was higher in the aged hippocampus. Thus, we can suggest that an age-related decrease in neurogenesis is not directly associated with place learning in aged rats.

## 1. Introduction

Aging is a complex process associated with multiple functional alterations and an increased probability of the appearance of pathological changes in various organs. Like all other organs, the aging brain undergoes changes, which are accompanied by a decrease in its functional capabilities, including cognitive functions; however, various cognitive domains may be differently affected. Aging significantly affects attention, perception, and memory in various individuals, whereas decision making, speech, and language remain relatively intact in older adults [1,2]. Furthermore, memory tasks requiring high levels of attention and controlled information processing are often difficult for the elderly, whereas declarative memory remains relatively unaffected [3]. In addition, the likelihood of neurodegenerative diseases increases with age [4,5], dictating the need for extensive studies of the age-related changes occurring in the brain and potentially underlying the development of age-related behavioral disturbances.

It is generally believed that the hippocampus is the most important structure of the brain involved in the processes of information encoding, consolidation, and recall [6,7]. In addition to its participation in cognitive processes, the hippocampus, along with other structures of the brain, regulates emotions, anxiety, and stress response [7,8,9,10,11]. The specific cytoarchitectonics of the hippocampus, the structure of neural networks, and the modular organization of functions due to its anatomy provide a high degree of plasticity in this part of the brain [12]. In addition, a unique feature of the hippocampus is neurogenesis, which persists in the subgranular zone (SGZ) of the dentate gyrus (DG) throughout life [13]. The reorganization of neural networks associated with learning and memory is based not only on the mechanisms of structural plasticity of existing neurons but probably requires the recruitment of newly formed nerve cells [14].

Neurogenesis in the mature mammalian brain is the process of the formation of new neurons during postnatal development. Due to the stem cell population preserved in the SGZ, the formation, migration, maturation, and incorporation of newly formed neurons are observed in the hippocampus [15]. Despite some evidence that neurogenesis in the hippocampus of higher mammals virtually halts at the early stages of postnatal ontogeny [16,17], other studies do not confirm this result [18]. In rodent brains, neurogenesis is observed throughout life, although it occurs less intensely in older animals than in young ones [19,20]. A progressive decrease in neurogenesis in rodents occurs between the 6th and 12th months of postnatal development, subsequently stabilizing at a very low level. Possible reasons for this include depletion of the stem cell pool in the hippocampus, a decrease in the number of signaling molecules in the stem cell microenvironment that stimulate the division and maturation of new neurons, and a decrease in the sensitivity of stem cells to signaling molecules [21]. Nitric oxide (NO) may be one of these regulators of cell proliferation and differentiation in the SGZ. Neurons expressing NO-synthase (nNOS) are located close to the germinative regions of the brain and send projections in the direction of dividing cells [22,23]. NO can play a dual role in the regulation of neurogenesis [24,25]. On the one hand, NOS inhibitors stimulate cell proliferation in the subventricular germinal zone, but not in the SGZ [26]. On the other hand, the administration of a NO donor stimulates neurogenesis both in the subventricular zone and in the SGZ [27], while knockout of the gene encoding nNOS promotes neurogenesis in both germinal regions of the brain [28,29,30]. These contradictions may be explained, in particular, by the fact that the effects of NO produced by different isoforms of NOS differ. 

In the present study, we aimed to investigate the changes in hippocampal neurogenesis in young mature and aged male Wistar rats subjected to training in a simplified water maze.

## 2. Results

### 2.1. Young Adult and Aged Rats Exhibit Similar Long-Term Habituation in the Open Field Test

Activity in the OFT includes motor and exploratory elements, among others. Repeated testing of animals in the same arena installed in the same conditions leads to decreases in ambulation and exploration (rearing), which are considered as habituation. Habituation in the tests performed with an interval of 24 h or longer may reflect the functioning of long-term memory [31,32]. In the present study, we compared activity of 4- and 24-month-old animals in the OFT with an interval of 25 days between the testing. In the OFT1, 24-month-old rats exhibited less square crossing as compared to 4-month-old animals (Figure 1a, Appendix A). This was supported by the data on RM ANOVA (Appendix A). No age-dependent effects on other indices of rat behavior were revealed, such as rearing, latency of exits out of the center, entries to the center, grooming, or defecation boli (Appendix A). We also did not observe differences in the behavior of rats of both groups during the second exposure to the same arena in the OFT2 (Figure 1a,b, Appendix A). However, significant decreases in the number of crossing and rearing were revealed in the OFT2 as compared to the OFT1 (Figure 1a,b, Appendix A), indicating long-term habituation in both groups of animals studied. This allows us to assume that long-term memory of environmental conditions consolidated normally and remained active in both 4- and 24-month-old rats.

### 2.2. Differences in Place Learning in Young Adult and Aged Rats Are Not Associated with Changes in Long-Term Memory

Two-day training in a simplified water maze showed that the rats of different age groups effectively learned to find a hidden platform (Figure 1c), although, in young adult rats, the latency decreased more rapidly on the first day and reached the plateau to trial 4. RM-ANOVA supported this observation: significant effects of the factor “age” *F*(1, 27) = 9.72, *p* < 0.01 and the factor “trial” *F*(4, 108) = 24.99; *p* < 0.001 were revealed during the first-day session; however, the “age” ×“trial” interaction *F*(4, 108) = 0.6; *p* > 0.6 was insignificant. Furthermore, there was no difference in the time spent in the target compartment of the maze by rats of different ages when performing a test trial without a platform at the end of the first day of training (Figure 1d).

In the first trial of the second training day, a slight increase in the platform search latency was observed compared to the last trial of the previous day, although this effect was not significant (*F*(1, 27) = 1.96; *p* = 0.17; Figure 1c) and did not depend on the age of the rats (*F*(1, 27) = 0.96; *p* = 0.37). During three trials on the second day, a decrease in the latency was observed in rats of both age groups (the “trial” factor (*F*(2, 54) = 3.25; *p* < 0.05). The effects of the “age” factor or the interaction between the factors were not found (*F*(1, 27) = 0.025; *p* > 0.87 and *F*(2, 54) = 0.32; *p* > 0.72, respectively). In a test trial without a platform at the end of the second training day, the time spent in the target compartment, where it was previously located, did not differ in either age group (Figure 1d). In the third test trial, which was carried out 48 h after training, young and old animals also spent similar time in the target compartment.

Thus, both 4- and 24-month-old animals learned the task in the simplified water maze, although the older group exhibited higher latency to search the platform as compared to the younger group. However, similar behavior of rats of various ages in the test trials indicates that different latencies of platform finding during the training session did not affect comparable efficiency of memory formation.

### 2.3. Aged Rats Exhibit Lower Exploration and Impaired Short-Term Memory in the Novel Object Recognition Test

The novel object recognition test (NORT) was applied to study short-term memory in young adult and aged rats. Total exploration time was reduced in aged animals as compared to young ones in both sessions (*p* < 0.05, M-W *U*-test). In addition, whereas young animals recognized a novel object in the test session (RI = 70 ± 7%, *p* < 0.05, single-sample *t*-test compared to 50% chance level), the aged animals failed to differentiate a novel object from the familiar one (RI = 59 ± 5%, *p* > 0.05, single-sample *t*-test compared to 50% chance level).

### 2.4. Neurogenesis Is Lower in the Hippocampus of Aged Rats 

To study hippocampal neurogenesis, the animals were injected with BrdU 28 days before the training in the water maze. After the end of testing, immunohistochemical detection of nuclei, which included BrdU, was performed. BrdU-containing nuclei were scarce in the SGZ of the DG in aged rats as compared to that in young animals (Figure 2a–d). The density of BrdU-positive nuclei in the SGZ of aged rats was almost eight times lower than that in the SGZ of young rats (*p* < 0.001, Mann–Whitney *U*-test; Figure 2e). In the CA4 field, the density of cells with BrdU-containing nuclei was similar in these two groups of rats (Figure 2e).

Since the number of BrdU-containing cells 1 month after the administration of this label does not reflect the level of proliferation of stem cells and/or neural precursors, which occurs at the moment of training or testing, we estimated the density of proliferating cells expressing PCNA in the DG (Figure 3). Similarly to BrdU-positive cells, the density of cells expressing PCNA was significantly lower in the hippocampus of the older group as compared to that of the younger one (*p* < 0.001; Figure 3c).

Finally, we estimated the differentiation of cells formed in the SGZ of the DG. The expression of microtubule-associated protein Dcx associated with the differentiation of cells via the neuronal pathway was used as a marker. In the hippocampus of young rats, Dcx-expressing cells looked like neurons with normally developed dendrites directed to the molecular layer of the DG (Figure 4a,c), whereas in aged rats, the dendrites were substantially less developed (Figure 4b,d). Moreover, double staining using the antibodies to Dcx and BrdU did not reveal any co-localization of proliferation and differentiation of neuronal precursor cells. In the groups of 4- and 24-month-old rats, BrdU-containing cells did not express Dcx and were located some distance from Dcx-expressing precursors. This was probably because the BrdU, included in the nuclei 1 month before the staining, could be found in either differentiating neurons or stem cells, which already finished or did not initiate differentiation. The density of Dcx-expressing cells was almost thirty times lower in the hippocampus of aged rats as compared to that of young animals (*p* < 0.001; Figure 3f).

Thus, the level of cell proliferation and neuronal-like differentiation was significantly inhibited in the hippocampus of aged rats.

### 2.5. The Expression of nNOS Selectively Changes in the Hippocampal Regions during Aging

Since NO may serve as a negative regulator of neurogenesis, we immunohistochemically assessed nNOS expression in the hippocampus of aged and young rats. Cells expressing nNOS were counted in the SGZ, CA4 field/hilus, and internal molecular layer, i.e., the areas located close to the DG. In the SGZ, nNOS-positive cells were located in the close vicinity to BrdU-containing cells (Figure 5a,b). The same was observed for co-localization of NOS with the other marker of proliferation PCNA (Figure 5c,d). Moreover, NOS-positive processes were in close vicinity of proliferating cells (Figure 5c,d; indicated by asterisk). The density of nNOS-positive neurons was significantly higher in the SGZ of aged rats as compared to that in young animals (*p* = 0.05; Figure 5e). In two other hippocampal subregions studied, the densities of nNOS-containing neurons did not differ. Thus, aging was associated with an increase in the density of nNOS-expressing neurons in the limited area of the hippocampus.

## 3. Discussion

In the present study, we demonstrated normal acquisition of long-term memory in aged rats because they exhibited normal habituation to novelty in the OFT and memory of the platform location in the water maze. Aged rats had higher latency during the training session of place learning in the water maze as compared to young rats. This difference was reduced to the end of the first day of the training and virtually disappeared on the second day. Importantly, throughout the test trials, both young and aged rats spent more than 30% of the test duration in the compartment of the maze where the platform was located during the training session. Thus, older animals learned the task as efficiently as younger rats. However, in general, aged rats demonstrated lower exploration as compared to young adults in the OFT and NORT. Moreover, their capability to recognize a novel object in the NORT was at the random level, in contrast to younger rats. 

Place learning in a water maze is one of the most common experimental paradigms used to study cognitive functions in aging animals. The authors of most studies assume that aging results in a deterioration in both the acquisition of the avoidance from the water and the development and recall of a memory trace of the location of the hidden platform [33,34,35,36]. However, the authors of some studies showed that the aging population contains both rats that learn this task like young animals, and rats with impaired learning and memory [37,38,39,40]. Noteworthy, these impairments can be detected only when regular test trials are introduced at the end of each training block and a special criterion based on the proximity of the search trajectory to the goal proposed by Gallagher et al. [41] was applied. Taken together, the data of the present study and the results reported by other groups show that aged animals can demonstrate satisfactory learning in a standard reference memory task in a water maze, although aged rats demonstrate a higher latency during training and/or less effective search strategies. This may be due to both the lower accuracy of the search and the lower activity of aged animals as compared to those of young ones. Importantly, studies in rodents [42,43,44] and humans [45] allowed Mulder et al. [44] to assume that aged individuals solve spatial tasks using a striatal-driven strategy instead of a hippocampal-driven strategy. This is probably because of the gradual loss in hippocampal function, and specifically aging-related loss in neurogenesis and spine-plasticity needed for memory formation [46,47,48]. Thus, spatial learning through exploratory navigation seems to be particularly vulnerable to the adverse effects of aging. However, even in humans, although senior individuals acquired spatial memories less accurately than young persons when they navigated in the environments, both groups did not differ in spatial learning performance when they viewed the environments from an aerial perspective [45].

Exploration of a novel environment promotes formation of memory on these conditions and stimulates the neural mechanisms responsible for memory functioning [49,50]. In the present study, we found that exploratory behavior in the aged rats was worse compared to that in the younger rats in two tests, specifically OFT and NORT. Lower exploratory activity did not prevent the formation of long-term memory, as detected using habituation in the OFT and a water maze place learning task, whereas it was crucial for short-term memory in the NORT. Chronic physical exercise maintains horizontal locomotor activity and significantly improves rearing activity, memory in the Morris water maze, and, notably, performance in the NORT in 32-month-old animals [51], probably because of its capability to stimulate neurogenesis in the DG [52,53]. We can suggest that memory disturbances observed in the present study may be associated with impaired neurogenesis, though this does not seem to be a causal relationship. There could be other factors contributing to the observed differences in task performance. 

Age-related decrease in neurogenesis is revealed in many mammalian species. In rodents, a main loss of the number of proliferating cells is observed at the age of 1.5–3 months [20,54], although in the SGZ, proliferation continues to decrease further. A large quantity of experimental data suggests that age-related slowing of neurogenesis in the mature brain is associated with deterioration in cognitive abilities [46,54,55,56,57]. However, most researchers note that aged animals can learn, including a spatial task in a water maze. Moreover, it has been shown that a high level of neurogenesis can predetermine a more effective strategy for finding a hidden platform in aging mice [54] or a shorter distance that rats swim during training [58]. The housing of aged animals in an enriched environment that enhances neurogenesis is usually accompanied by an improvement in learning and memory to the level of young ones [54,56]. Interestingly, in adult animals, inhibition of neurogenesis by ionizing radiation does not interfere with learning in water [59] and T- [60] mazes. This may indicate that substantial neurogenesis in the mature hippocampus is not necessary for the formation of spatial memory at any age. Moreover, data on the direct effect of learning in a water maze on the intensity of neurogenesis in the SGZ reported by different groups [61,62] are quite contradictory.

Neurogenesis in the mature brain is controlled by a variety of internal and external factors, including neurotrophins, transcription factors, and cell cycle proteins, which regulate stem cell proliferation and differentiation of newly formed cells, as well as maintain a specific state of the neurogenic niche [13,24,63,64,65]. The formation and maintenance of a neurogenic niche that promotes the proliferation of stem cells and the determination of their future fate depend on the normal functioning of the immune system and microvascular blood flow [66]. Regulatory effects of cells located in the close vicinity of proliferative zones can be mediated by a variety of secreted molecules. These signaling molecules include NO, which is synthesized by the nNOS of GABAergic neurons located in the Ammon horn and DG [67,68], and even some PSA-NCAM-expressing neuronal precursors [69]. It is known that during aging there is a significant increase in NO synthesis, which is most pronounced in the neocortex, hypothalamus, and cerebellum of aging animals [70,71,72,73], but not in the hippocampus. In the current study, it was found that brain aging was accompanied by an increase in the expression of nNOS in the SGZ, i.e., precisely near the proliferative zone of the hippocampus. This is consistent with the previously reported data showing that the expression and activity of various NOS isoforms differ significantly in specific parts of the hippocampus [74]. In particular, this group showed that, in aged rats, the maximum increase in NOS activity occurred in the DG. Taken together, these data suggest that a local increase in NO production in older animals may be among the causes of neurogenesis inhibition. It was mentioned above that NO inhibits neurogenesis under normal physiological conditions [26,28,75]. Moreover, knockout of the nNOS encoding gene results in an increase in cell proliferation in the SGZ of 2-month-old mice but a decrease in the stem cell population in the DG and cell proliferation in the SGZ of 18-month-old mice [28,76]. The exact mechanisms of this effect remain obscure. NO may inhibit proliferation partially by inhibiting the epidermal growth factor (EGF) receptor tyrosine kinase and Akt phosphorylation, as shown in neurospheres derived from the subventricular zone [76]. The inhibitory effect of NO may also be mediated by peroxynitrite, which is formed as a result of the interaction of NO and the superoxide anion radical [30]. Peroxynitrite activates signaling cascades associated with non-receptor Src kinases, which regulate the processes of proliferation, differentiation, and migration of many cell types, including stem cells located in germinal regions of the brain [77]. However, an opposite, stimulatory effect of NO has been also reported. A long-term administration of a NO donor to the aged mice with metabolic syndrome stimulated the accumulation of BrdU in the hippocampus of the animals [78]; however, no generation of new neurons was presented in the study. Moreover, this increase in the accumulation of BrdU did not correlate with a significant improvement in memory acquisition in the Morris water maze. This might be associated with an indirect effect of NO since the NO donor decreased blood glucose promoting conditions stimulating proliferation of stem cells [79] and adult neurogenesis [80]. 

## 4. Materials and Methods

### 4.1. Animals

Male Wistar rats (Stolbovaya Breeding Center, Moscow region, Russia) were housed as five animals per a cage in the institutional colony room and maintained with free access to food and water at a light cycle of 08.00–20.00. These were 3- and 23-month-old animals at the start of the experiment (*N* = 15 and *N* = 14, respectively).

### 4.2. 5-Bromo-2′-deoxyuridine (BrdU) Injection

After adaptation to the housing conditions, the rats were shortly handled for 3 days. Then, the animals were injected with BrdU solution at a dose of 50 mg/kg in the volume of 1 mL/kg for 5 days twice a day. BrdU (MP Biomedicals, Irvine, CA, USA) was dissolved in 0.008 N NaOH immediately before injection.

### 4.3. Open Field Test

In order to study animal activity and habituation, the open field test (OFT) was applied. For this purpose, the rats were individually placed in a white ound arena with a diameter of 120 cm surrounded with 40 cm high wall. The arena was situated in the center of a quiet room and lit with four 20 W luminescent bulbs located 2 m over the center of the arena. The floor of the arena was divided into squares 20 × 20 cm in size. The animal was placed in the center of the arena and allowed to move freely. The following indices were manually recorded: latency to exit out of the center, number of square crossings, rearing, number of entries to the center, grooming, and number of defecation boli. In order to study long-term habituation, the animals were tested in the same arena under the same conditions 25 days later. Habituation was estimated as a decrease in the number of total crossings and rearing counted within the test duration. 

### 4.4. Novel Object Recognition Test

The novel object recognition test (NORT) was performed to assess the short-term memory in young and aged rats [81]. On the day before the test, the animals were allowed to freely explore the empty experimental cage (white plastic box 60 × 35 × 20 cm) for 5 min. On the next day, each animal was placed in the cage, and two identical objects were presented for 3 min (training session) in two opposite corners of the cage. During the training session, the animal was allowed to explore the cage and the objects, and its behavior was video recorded. Then, the animal was placed back in its home cage for 30 min. During the test session, the animal was returned to the experimental cage with two different objects, one of them being identical to objects presented in the training session, and another one being a new one (with left or right object changed randomly), and again allowed to explore the cage and objects for 3 min. The video recording was manually analyzed to measure the time spent to explore each object (both during the training session and the test session). The total exploration time was measured in the training session, and the ratio between the novel object exploration time and total exploration time was calculated in the test session as the recognition index (RI).

### 4.5. Water Maze Training

On day 28 after the start of the experiment, the rats were trained to find a hind platform in a simplified water maze task [82]. The water maze was a rectangular pool (100 × 60 × 35 cm) filled with water to a depth of 27 cm. The water temperature was 23 ± 1 °C. The pool was divided into three equal parts. In the center of one lateral part, a clear Perspex platform with a diameter of 15 cm was located and hidden 1 cm beneath the water surface. Skimmed milk was added to the water for turbidity. Laboratory interior items served as extra maze cues. The rats were trained to search for a platform for two days. On the first day, 5 trials were conducted. Each trial was started by placing the rat in the northern or southern corners, located in the part of the maze farthest from the platform, with its nose to the wall of the pool. The rat had to find the platform in 60 s. If she found a platform, she was left on the platform for 20 s. If the rat did not find the platform, then she was gently pushed towards the platform until the rat climbed onto it. In the latter case, the rat was also left on the platform for 20 s, and the latency was counted as 65 s. On the second day of training, the rats were given 3 trials. At the end of each trial, the rat was taken out of the pool, dried with a towel and under a warm air flow, and returned to the home cage. The latency of finding the platform was used as an indicator of learning. The starting position for each animal was chosen randomly and was not changed during the entire training period. At the end of each training day, a test trial was carried out 60 min after the end of the training session. The third test trial was carried out 48 h after the second test. For the test trial, the platform was removed from the pool and the rat was allowed to search for the platform for 30 s (or 60 s in the third test trial); the time spent by the rat in the part of the maze where the platform was located during training was recorded.

### 4.6. Immunohistochemistry

For immunohistochemical studies, randomly selected rats aged 4 (*N* = 8) and 24 months (*N* = 8) were overdosed with chloral hydrate (400 mg/kg). The brain was fixed by intracardial perfusion of 4% paraformaldehyde in 0.1 M phosphate buffer (pH 7.4). At the end of perfusion, the brain was removed and fixed in the same fixative for at least 48 h. Coronal sections of the brain with a thickness of 50 um were prepared using a Leica 1200S vibrating microtome (Leica Microsystems, Wetzlar, Germany). The sections were placed in a cryoprotective liquid and stored at −18 °C. To stain the sections, antigen retrieval was performed in 0.01 M citrate buffer (pH 6.0) for 10 min at 95 °C. DNA was denatured in 2 M HCl solution for 30 min at 37 °C. The sections were treated with 0.1 M borate buffer (pH 8.0) for acid neutralization. Incubation in 3% H_2_O_2_ solution for 10 min was used for quenching endogenous peroxidases. After triple washing with 0.3% triton X-100 (Serva, Heidelberg, Germany) in 0.01 M phosphate-salt buffer (PBST), nonspecific binding was blocked by 1 h incubation in 5% nonimmune goat serum solution (MP Biomedicals, USA) in PBST. After that, a solution of mouse monoclonal antibodies to BrdU (BD Pharmingen, San Diego, CA, USA) diluted with a blocking buffer in a ratio of 1:4000 was applied to the sections. The sections were incubated for 18 h at 4 °C. After that, the sections were washed in PBST and incubated with biotinylated goat antibodies to mouse IgG (Jackson ImmunoResearch Laboratories, West Grove, PA, USA), diluted with a blocking buffer in a ratio of 1:800, for 2 h at room temperature. After washing, the sections were treated with a solution of avidin-biotin complex conjugated with horseradish peroxidase (Vectastain ABC kit, Vector Labs, Burlingame, CA, USA) for 1 h. 3,3′-diaminobenzidine (Sigma FAST kit, Sigma-Aldrich, St. Louis, MO, USA) was used as a chromogen. At the end of the procedure, the sections were mounted on gelatin-coated slides, dehydrated in ethanol solutions, cleared in xylene, and coverslipped in a Histofluid medium (Marienfeld, Lauda-Königshofen, Germany). The same protocol was applied for the detection of proliferating cell nuclear antigen (PCNA) except for the DNA denaturing step. Mouse monoclonal anti-PCNA antibody (Santa Cruz Biotechnology, Dallas, TX, USA) was used in a dilution of 1:1000.

Immunohistochemical evaluation of nNOS and doublecortin (Dcx) expression was conducted using a similar protocol, except for the antigen retrieval and DNA denaturing steps. Polyclonal rabbit antibody to nNOS (Millipore, Burlington, MA, USA) was used in a 1:500 dilution. Polyclonal rabbit anti-Dcx antibody (Abcam, Waltham, MA, USA) was used in a 1:400 dilution.

Double immunofluorescent staining was used for the experiments with co-localization of Dcx or nNOS and BrdU-containing neurons. The sections were washed with PBST and consequently incubated in 2 M HCl (30 min at 37 °C), 0.1 M borate buffer (10 min at 25 °C). Unspecific binding was blocked by 1 h incubation in the blocking buffer, containing 5% nonimmune goat serum in PBST. Then, the sections were incubated in a mixture of rabbit anti-Dcx antibody (Abcam, Waltham, MA, USA) diluted in a ratio of 1:400 or rabbit anti-nNOS (Millipore, Burlington, MA, USA), diluted in a ratio of 1:500, and anti-BrdU (BD Pharmingen, San Diego, CA, USA) diluted in a ratio of 1:4000, antibodies. After overnight incubation with primary antibodies, the sections were washed in PBST and incubated with a mixture of goat anti-rabbit IgG antibody conjugated with Alexa-488 and goat anti-mouse IgG antibody conjugated with Alexa-546 (both Invitrogen, Waltham, MA, USA) for 3 h in a 1:500 dilution, or a mixture of goat anti-mouse IgG antibody conjugated with Alexa-488 and goat anti-rabbit IgG antibody conjugated with Alexa-546 (both Invitrogen, Waltham, MA, USA) for 3 h in a 1:500 dilution. After additional washing with PBS, the sections were mounted onto the slides and coverslipped with Fluoroshield with DAPI anti-fade (Invitrogen, Waltham, MA, USA). Fluorescent imaging was performed using an Imager.Z2 equipped with Apotome2 pseudoconfocal microscope and AxioCam MRm camera (both Carl Zeiss, Gina, Germany) and the filter sets FS 49 (DAPI EX BP G365. BS 395: EM BP 445/50), FS 09 (Alexa Fluor 488 EX BP 450-490 FT510 EM LP520), and FS 15 (Alexa Fluor 546 EX BP 546/12 FT 580 EM LP 590). Images were captured and processed using ZEN 2011 software (Carl Zeiss, Gina, Germany).

### 4.7. Cell Count

BrdU-positive cells were counted in the DG SGZ (a band having a width of two cell body diameters inside the granule cell layer [15]) and the CA4 region, defined as the area located between the branches of the DG. The calculation was carried out on 10 sections of the hippocampus located at a distance of 600 um from each other. The first section in the series was chosen by a random unbiased sampling scheme. Cells were counted at magnification ×400 using a Leica DM6000 B microscope (Leica Microsystems, Wetzlar, Germany). Densely colored BrdU-containing nuclei and transparent nuclei with many colored points and clearly visible boundaries of the nucleus were counted in one focal plane. The area of SGZ and CA4 was measured using FIJI software [83]. The number of cells was averaged and represented as the relative density of cells per mm^3^. 

### 4.8. Statistical Analysis

The correspondence of data to the normal distribution was evaluated using the Shapiro–Wilk criterion. Latency changes in the learning process were evaluated using repeated-measure analysis of variances (RM-ANOVA) with the factors “age” and “trial”. Evaluation of behavior in test trials and comparison of the number of cells was carried out using Student’s *t*-test. Behavior in the OFT1 and OFT2 was estimated using RM-ANOVA with the factors “age” and “test” followed by the Tukey HST for unequal N post hoc comparison. Behavior in the NORT was estimated using the Mann–Whitney *U*-test and single-sample *t*-test. The Mann–Whitney *U*-test was applied to immunohistochemical data. The differences were considered significant at *p* < 0.05.

## 5. Conclusions

In the present study, a significant decrease in neurogenesis was confirmed in aged rats as compared to young animals. All stages of neurogenesis were impaired, including proliferation, differentiation, and, presumably, incorporation of newborn neurons in the DG. Increased production of NO due to the enhanced expression of nNOS in the close vicinity of the hippocampal proliferating zone may cause such a decrease in the generation of new neurons. Significant inhibition of neurogenesis in aged rats might be associated with moderate learning disturbances observed during the water maze training; however, aged rats were able to perform this task and exhibited memory capacity comparable with that of young rats. These data support the idea that, under some conditions, aged animals can exhibit cognitive abilities with only minor changes or even when unimpaired. Furthermore, in the aged group, learning and memory were probably not closely related to the level of current neurogenesis. We believe that further studies of age-related alterations of neurogenesis will allow us to reveal the importance of this process for cognitive functions during aging.

## 6. Limitations of the Study

Our study has several limitations. First, our study was focused on aged male Wistar rats, which may not fully represent the complexity of age-related cognitive changes in humans or other animal models. Therefore, the findings should be generalized with caution to other species or even other strains of rats. Second, we applied a simplified water maze to study learning and memory. The smaller size of this pool and easier identification of the target sector significantly limit the number of behavioral indices used to describe the progress of animal behavior, and the latency to find the hidden platform is the major index. Despite the fact that the use of a water maze task to assess memory acquisition is a common approach, it has its own limitations. Behavior in the water maze allows the assessment of spatial memory, and the findings might not fully translate to other types of memory or cognitive functions affected by hippocampal neurogenesis. Our study was focused on a specific aspect of memory acquisition and learning, whereas other cognitive functions might also be influenced by hippocampal neurogenesis. Memory acquisition is just one component of cognitive function, and a broader range of tests should be applied to provide a more comprehensive picture. Third, we considered the data from all test trials performed 1 h after the end of the training session (test 1 and 2) or 48 h after the last training session (test 3) as the indices of long-term memory, although behavior of rats during test trials 1 and 2 may reflect conditions of both short-term and long-term memory. Moreover, our behavioral protocol did not include extensive behavioral testing or training since this type of animal treatment might affect neurogenesis per se. Fourth, in this study, we did not specifically address the alterations in stem cell population. Furthermore, the methods used to measure neurogenesis in the present study have their own limitations in accuracy and sensitivity. Variability in these measurements, as well as in protocols of BrdU application, could impact the reliability of the conclusions. There could be other factors contributing to the observed differences in task performance. Fifth, our study proposes a potential link between increased nNOS expression and decreased neurogenesis, but it does not fully explore the underlying molecular mechanisms or interactions involved in this process. For example, we did not expect to modify nNOS activity in rats in vivo since there are no available nNOS inhibitors specifically affecting this enzyme in the brain after systemic administration. 

## Figures and Tables

**Figure 1 ijms-24-13276-f001:**
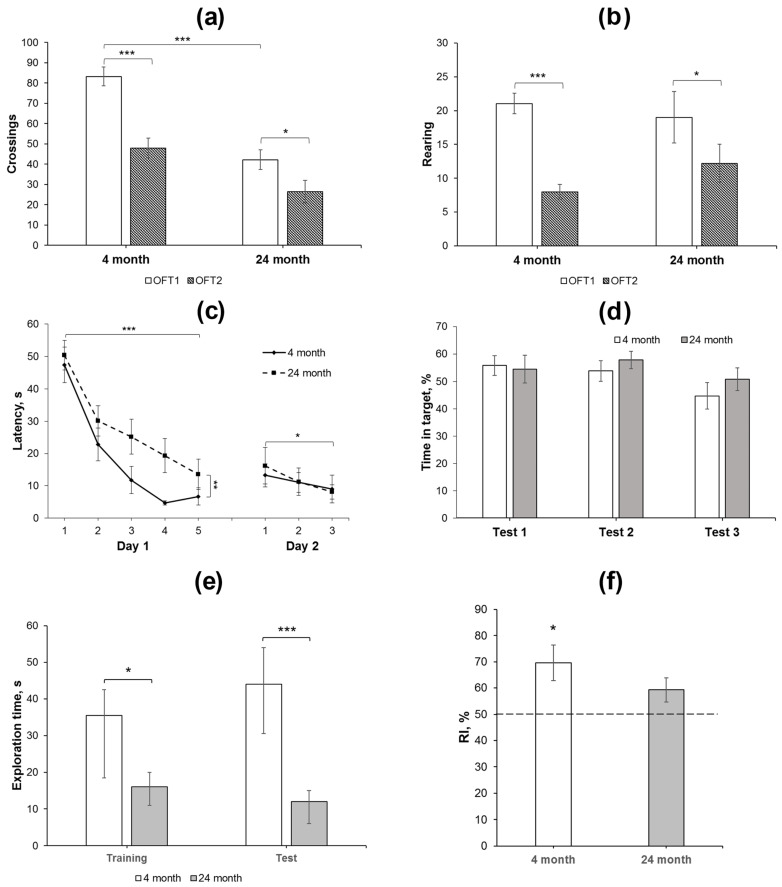
Behavioral analysis of changes in long-term and short-term memory in young adult and aged rats. (**a**,**b**) Square crossing and rearing, respectively, as the indices of long-term habituation in the OFT1 and 2 in 4- (*N* = 15) and 24-month-old (*N* = 14) Wistar rats. The differences are significant at ***—*p* < 0.001; *—*p* < 0.05 according to ANOVA with repeated measures (see in the text and in Appendix A). (**c**,**d**) Water maze training: (**c**) changes in the latency to find a platform during 2-day training and (**d**) time spent in the target compartment during the test trial 1 (end of day 1 training), test 2 (end of day 2 training), and test 3 (48 h after the last training day). The differences are significant at ***—*p* < 0.001; **—*p* < 0.01; *—*p* < 0.05 according to ANOVA with repeated measures (see in the text). (**e**,**f**), NORT performance in young adult (*N* = 8) and aged (*N* = 7) animals. The test session with the novel object was performed 30 min after presentation of similar objects in the probe session. (**e**) Total time spent for exploration of both objects was reduced in old animals as compared to young animals in the training and test sessions. The differences are significant at ***—*p* < 0.005; *—*p* < 0.05 according to M-W *U*-test; (**f**) recognition index (RI), calculated as T_new_ × 100/(T_old_ + T_new_), was significantly higher than the 50% chance level in young animals, but not in aged ones. The differences are significant at *—*p* < 0.05 according to single-sample *t*-test. Data in (**a**–**d**,**f**) are presented as M ± sem and in (**e**) as median ± (LQ − UQ).

**Figure 2 ijms-24-13276-f002:**
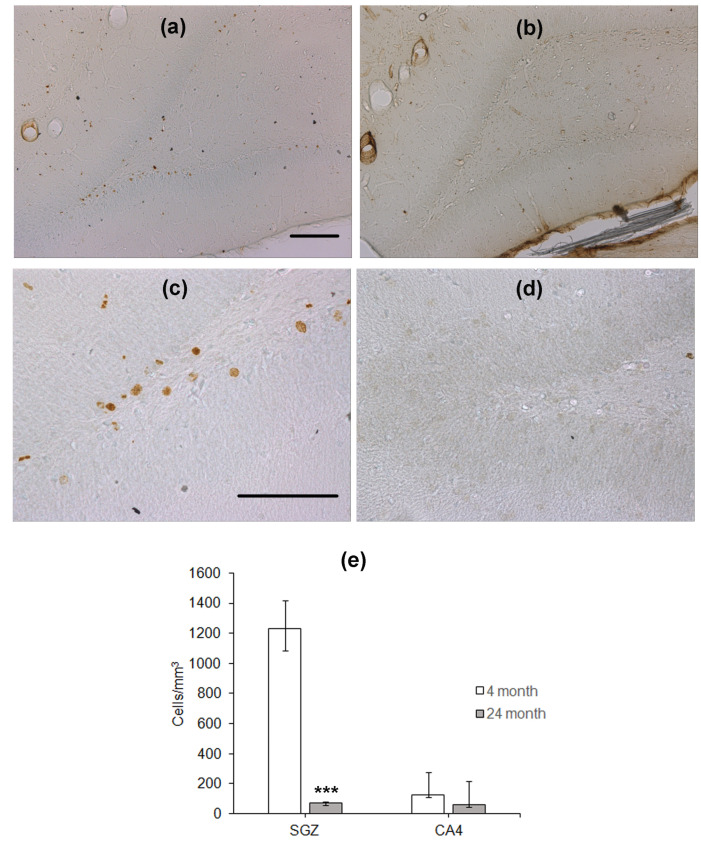
Immunohistochemical detection of BrdU-containing nuclei in the SGZ of 4- (**a**,**c**) and 24-month-old (**b**,**d**) Wistar rats. Scale bar—200 µm. (**e**) Density of BrdU-containing neurons in the hippocampal SGZ and CA4 area. Data are presented as median (upper–lower quartile). *N* = 8. The differences are significant at ***—*p* < 0.001 according to Mann–Whitney *U*-test.

**Figure 3 ijms-24-13276-f003:**
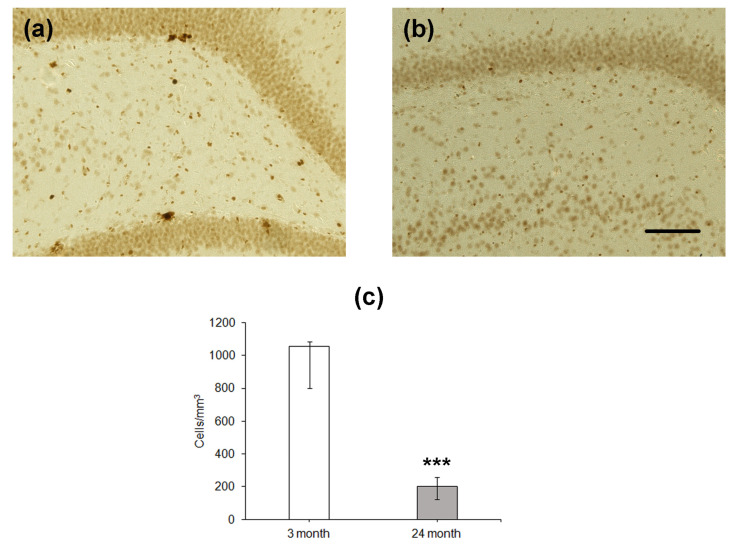
Immunohistochemical detection of PCNA-expressing neurons in the hippocampal DG of 4- (**a**) and 24-month-old (**b**) Wistar rats. Scale bar—200 µm. Density of PCNA-containing (**c**) neurons in the hippocampal SGZ. Data are presented as median (upper–lower quartile). *N* = 8. The differences are significant at ***—*p* < 0.001 according to Mann–Whitney *U*-test.

**Figure 4 ijms-24-13276-f004:**
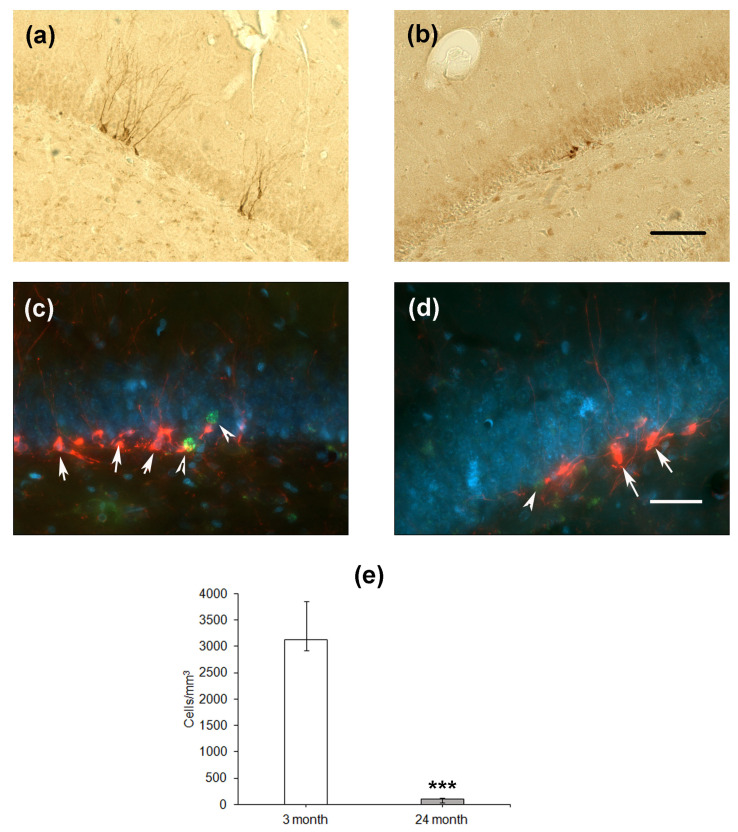
Immunohistochemical detection of Dcx-expressing neuronal precursors in the hippocampal DG of 4- (**a**) and 24-month-old (**b**) Wistar rats. Scale bar—200 µm. Double immunostaining of Dcx-expressing neuronal precursors (arrows) and BrdU-containing nuclei (arrowheads) in the DG of 4- (**c**) and 24-month-old (**d**) rats. Scale bar—50 µm. The sections were counterstained with DAPI to visualize cellular nuclei. Density of Dcx-positive (**e**) neuronal precursors in the SGZ. Data are presented as median (upper–lower quartile). *N* = 8. The differences are significant at ***—*p* < 0.001 according to Mann–Whitney *U*-test.

**Figure 5 ijms-24-13276-f005:**
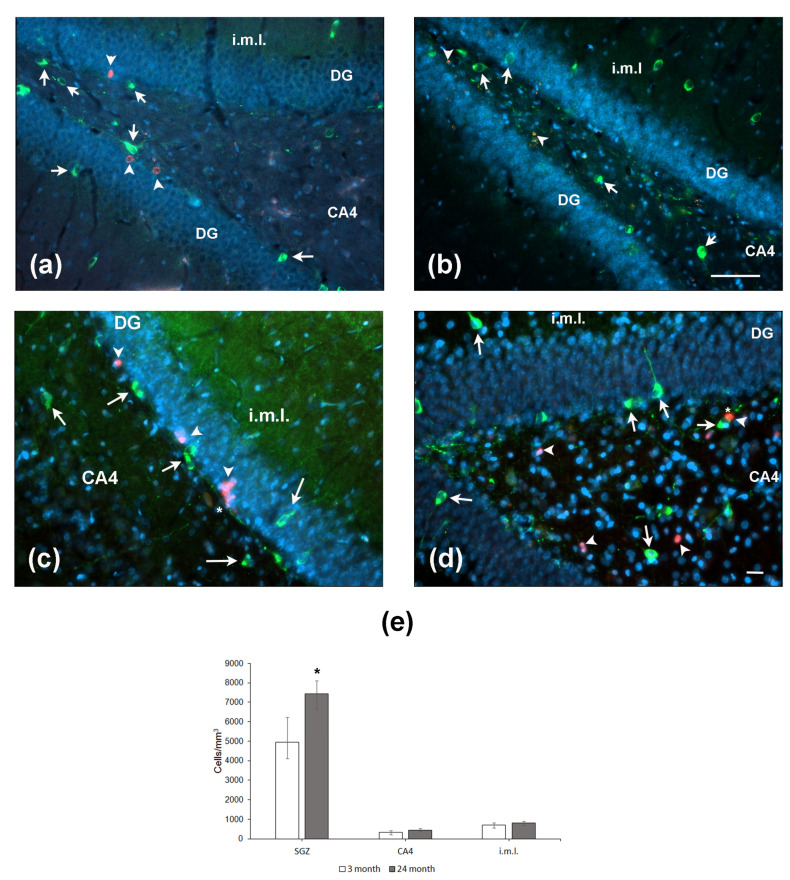
Expression of nNOS and markers of cell proliferation BrdU (**a**,**b**) and PCNA (**c**,**d**) in the hippocampus of 4- (**a**,**c**) and 24-month-old (**b**,**d**) Wistar rats. (**a**,**b**) Double immunofluorescent detection of BrdU (red, arrowheads) and nNOS (green, arrows) containing neurons in the DG, CA4, and internal molecular layer (i.m.l.) of 4-mo (**a**) and 24-mo (**b**) rats. Scale bar—200 µm. (**c**,**d**) Double immunofluorescent detection of PCNA (red, arrowheads) and nNOS (green, arrows) containing neurons in the hippocampus of 4-mo (**c**) and 24-mo (**d**) rats. Scale bar—50 µm. Asterisk indicates the NOS+ processes in close vicinity to (**c**) or over (**d**) the PCNA-expressing cells. DAPI was used to stain cell nuclei. (**e**) Density of nNOS-containing neurons in the hippocampal SGZ, CA4, and i.m.l. Data are presented as median (upper–lower quartile). *N* = 8. The differences are significant at *—*p* < 0.05 according to Mann–Whitney *U*-test.

## Data Availability

The datasets generated during and/or analyzed during the current study are not publicly available due to the requirements of the Institute of Higher Nervous Activity and Neurophysiology, but are available from the corresponding author on reasonable request.

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
