# Peer review of "Decreased Hippocampal Neurogenesis in Aged Male Wistar Rats Is Not Associated with Memory Acquisition in a Water Maze"

_ijms, 2023, doi:10.3390/ijms241713276_

Round 1

Reviewer 1 Report (Previous Reviewer 2)

The authors substantially improved their resubmitted manuscript. I have no particular criticism. The current manuscript can be published as it stands.

Author Response

We thank the Reviewer 1 for the repeated estimation of our study.

Reviewer 2 Report (New Reviewer)

The Ms submitted is acceptable for publication on IJMS provided that the minor point described below will be addressed.

Minor point

The submitted version of the Ms still presented several parts of the text highlighted in yellow - this must be rectified.

Author Response

We thank the Reviewer 2 for the helpful comments. New changes highlighted in yellow were included according the criticism of the other Reviewers.

Reviewer 3 Report (New Reviewer)

The manuscript titled "Decreased hippocampal neurogenesis in aged male Wistar rats is not associated with memory acquisition in a water maze" investigates the relationship between age-related changes in hippocampal neurogenesis and memory acquisition in a water maze task in male Wistar rats. While the study provides interesting insights, there are several potential downsides or limitations that should be considered:

1.    The study focuses on aged male Wistar rats, which may not fully represent the complexity of age-related cognitive changes in humans or other animal models. The findings might not generalize to other species or even other strains of rats.

2.    The use of a water maze task to assess memory acquisition is a common approach, but it has its limitations. The water maze assesses spatial memory, and the findings might not fully translate to other types of memory or cognitive functions affected by hippocampal neurogenesis. The study focuses on a specific aspect of memory acquisition and learning, neglecting other cognitive functions that might be influenced by hippocampal neurogenesis. Memory acquisition is just one component of cognitive function, and a broader range of tests could provide a more comprehensive picture.

3.    The methods used to measure neurogenesis (PCNA-positive, Dсх-positive, and BrdU-positive cell counts) have limitations in accuracy and sensitivity. Variability in these measurements could impact the reliability of the conclusions.

4.    While the study suggests that decreased neurogenesis is not directly associated with memory acquisition, there could be alternative mechanisms or compensatory processes that contribute to the observed cognitive performance in aged rats. The study proposes a potential link between increased neuronal NO-synthase expression and decreased neurogenesis, but it does not fully explore the underlying molecular mechanisms or interactions involved in this process.

5.    The study identifies a correlation between decreased hippocampal neurogenesis and age-related changes in the water maze task but does not establish a causal relationship. There could be other factors contributing to the observed differences in task performance.

It's important to consider these downsides and limitations when interpreting the results of the study and drawing broader conclusions about the relationship between hippocampal neurogenesis and age-related cognitive changes.

Author Response

Firstly, we want to thank the Reviewer 3 for a definite wording of several limitations of our study. These limitations were addressed in the previous version of the manuscript in a special chapter (paragraph 6), though only partially. We tried to do our best to discuss these limitations in the new version of the manuscript (all changes are highlighted in yellow).

The manuscript titled "Decreased hippocampal neurogenesis in aged male Wistar rats is not associated with memory acquisition in a water maze" investigates the relationship between age-related changes in hippocampal neurogenesis and memory acquisition in a water maze task in male Wistar rats. While the study provides interesting insights, there are several potential downsides or limitations that should be considered:

  1. The study focuses on aged male Wistar rats, which may not fully represent the complexity of age-related cognitive changes in humans or other animal models. The findings might not generalize to other species or even other strains of rats.

We agree with this opinion. However, this concerns the majority of experiments on animal models: it is difficult to translate most animal studies in aging to human population because of significant evolutionary gap between the species. We included this concern in paragraph 6 (Lines 508-511): “First, our study was focused on aged male Wistar rats, which may not fully represent the complexity of age-related cognitive changes in humans or other animal models. Therefore, the findings should be generalized with caution to other species or even other strains of rats.”    

  1. The use of a water maze task to assess memory acquisition is a common approach, but it has its limitations. The water maze assesses spatial memory, and the findings might not fully translate to other types of memory or cognitive functions affected by hippocampal neurogenesis. The study focuses on a specific aspect of memory acquisition and learning, neglecting other cognitive functions that might be influenced by hippocampal neurogenesis. Memory acquisition is just one component of cognitive function, and a broader range of tests could provide a more comprehensive picture.

All behavioral approaches have their specific limitations in studies on cognitive functions in animals. That's why, in the present study, we applied three different task to study learning and memory in rats, such as water maze, habituation in the open field, and novel object recognition. We understand that each of them allows to study different aspects of memory acquisition, storage, and recall, but also exploration and attention in any case. Therefore, some other components of cognitive functions were also studied. We agree that a broader range of tests is better than the three used, but it is not possible to do all possible studies on the same animal cohort. We also discussed this concern (paragraph 6 (Lines 515-520): “Despite the fact that the use of a water maze task to assess memory acquisition is a common approach, it has its own limitations. Behavior in the water maze allows to assess spatial memory, and the findings might not fully translate to other types of memory or cognitive functions affected by hippocampal neurogenesis. Our study was focused on a specific aspect of memory acquisition and learning, whereas other cognitive functions might also be influenced by hippocampal neurogenesis.”  

  1. The methods used to measure neurogenesis (PCNA-positive, Dсх-positive, and BrdU-positive cell counts) have limitations in accuracy and sensitivity. Variability in these measurements could impact the reliability of the conclusions.

We fully agree that all the indices used have definite limitations in terms of accuracy and sensitivity. Moreover, even the protocol of BrdU administration may significantly affect the results. Therefore, we used three different methods estimating different aspects of neurogenesis. This is also discussed in paragraph 6 (Lines 528-532): “Furthermore, the methods used to measure neurogenesis in the present study have their own limitations in accuracy and sensitivity. Variability in these measurements as well as in protocols of BrdU application could impact the reliability of the conclusions. There could be other factors contributing to the observed differences in task performance.”

  1. While the study suggests that decreased neurogenesis is not directly associated with memory acquisition, there could be alternative mechanisms or compensatory processes that contribute to the observed cognitive performance in aged rats. The study proposes a potential link between increased neuronal NO-synthase expression and decreased neurogenesis, but it does not fully explore the underlying molecular mechanisms or interactions involved in this process.

This point is discussed in Lines 531-535: “There could be other factors contributing to the observed differences in task performance. Fifth, our study proposes a potential link between increased nNOS expression and decreased neurogenesis, but it does not fully explore the underlying molecular mechanisms or interactions involved in this process. For example, we did not expect to modify nNOS activity in rats in vivo since there are no available nNOS inhibitors specifically affecting this enzyme in the brain after systemic administration.”

  1. The study identifies a correlation between decreased hippocampal neurogenesis and age-related changes in the water maze task but does not establish a causal relationship. There could be other factors contributing to the observed differences in task performance.

We included this concern into the discussion (Lines 296-299): “We can suggest that memory disturbances observed in the present study may be associated with impaired neurogenesis, but this does not seem to be a causal relationship. There could be other factors contributing to the observed differences in task performance.”

It's important to consider these downsides and limitations when interpreting the results of the study and drawing broader conclusions about the relationship between hippocampal neurogenesis and age-related cognitive changes.

We have taken into account all these problems and drawbacks, included respective information into the text and hope that it significantly improved the manuscript. We greatly appreciate the recommendations of Reviewer 3.

Round 2

Reviewer 3 Report (New Reviewer)

The authors have addressed the concerns raised to a certain extent, hence the manuscript can be accepted in present form.

This manuscript is a resubmission of an earlier submission. The following is a list of the peer review reports and author responses from that submission.

Round 1

Reviewer 1 Report

In the manuscript entitled ‘Decreased hippocampal neurogenesis in aged male Wistar rats is not associated with memory acquisition in a water maze’, Stepanichev et al. investigated whether age-related changes in hippocampal neurogenesis are associated with learning and memory disturbances. They claimed that the reduced hippocampal neurogenesis in aged rats is not associated with spatial learning and memory. I have the following concerns:

1.      In Figure 1, the authors state that differences in spatial learning between 24- and 48-month-old rats are not associated with changes in long-term memory. However, there are no long-term memory tests at all. In addition, I am wondering whether there is a real difference in the first latency to locate the platform on the second training day. As for probe trials, the first latency to target and the number of target crossings should be provided. Additional behavioral assays (such as Barnes maze) would be necessary to make a sound conclusion.

2.      In Figures 2 and 3, the co-localization assay of BrdU and PCNA/Dcx is a better method to analyze the proliferation and differentiation of NSPCs.

3.      In Figure 4, I or most readers would like to know the relationship between nNOS+ cells and NSPCs. Please provide high magnification confocal images demonstrating the identity of nNOS+ cells.

4.      The authors hypothesized that the declined neurogenesis in aged hippocampus is related to increased local production of nitric oxide. Unfortunately, the authors did not test such an interesting hypothesis.

Minor editing of English language required.

Author Response

The authors greatly appreciate the criticism of the Reviewer and have addressed all the concerns and revised the manuscript accordingly. The chapter Limitations of the study has been added to address drawback highlighted by the Reviewer. All modifications are highlighted in yellow. 

  1. In Figure 1, the authors state that differences in spatial learning between 24- and 48-month-old rats are not associated with changes in long-term memory. However, there are no long-term memory tests at all. In addition, I am wondering whether there is a real difference in the first latency to locate the platform on the second training day. As for probe trials, the first latency to target and the number of target crossings should be provided. Additional behavioral assays (such as Barnes maze) would be necessary to make a sound conclusion.

We did not use the group of 48-month-old animals for the study, the 4- and 24-month-old rats were studied. There are some experimental problems with practical discrimination between short-term and long-term memory. In our studies, we believe that all test trials may represent the retrieval of information from long-term memory since they were performed either 1 h (Test 1 and Test 2) after the end of each acquisition session or 48 h after the last training trial of Day 2. Thus, we used the approach to test long-term memory according to the options within the protocol applied. On Day 2 of training a statistically significant decline was observed between the trial 1 and trial 3 of the session according to repeated measures ANOVA. This is indicated in Fig 1a by asterisk and respective description is presented in the text. No difference between 4- and 24-month-old groups was revealed.

Since the size of the simplified pool used in our study is substantially smaller as compared to the standard pool employed for the Morris water maze task, the animal identified the target sector easier and spent there most of the test time. Importantly, every turn of the body and swim from   one wall to another wall in this sector of the pool will inevitably cross the platform location. Therefore, there was no reason to estimate crossings the platform location site as more representative feature of memory. All animals left the start sector of the pool immediately after the placing in the pool and performed their first visit to the target sector. In case rats were unable to find the platform, they searched mostly within the target sector. Thus, the latency to the first visit to the target sector during the test trial in this simplified version of the maze is not informative.

Since these experimental points are important for the interpretation of the data, they are declared in the chapter Limitations of the study in the revised version of the paper.

We agree that several memory tests are better than the use only one. However, it is tricky to develop a protocol which include many tests for the same animals; we should bear in mind that    the tests for memory acquisition and retrieval may affect neurogenesis per se. It is impossible to perform a new series of experiments on the animals included in this experiment. This is an unavoidable limitation of this study. However, we are now performing the experiment in aging 14-month-old rats which includes training them in the Barnes maze in. Remarkably, our preliminary data show that these rats are able to perform finding of the safety box in this maze, though less efficiently as younger rats.

  1. In Figures 2 and 3, the co-localization assay of BrdU and PCNA/Dcx is a better method to analyze the proliferation and differentiation of NSPCs.

We agree with Reviewer’s opinion that colocalization of several markers is the best way to make a conclusion on the cell fate. However, in some cases it is difficult, if possible at all, to do this, and this is an inevitable limitation. For example, mouse anti-PCNA antibody applied in our study requires heat antigen retrieval, which is critical for Dcx staining because Dcx poorly survives this pretreatment. Therefore, we only performed co-staining for BrdU and Dcx. Microphotographs are presented in corrected Fig. 4. It can be clearly seen that BrdU-positive nuclei were not colocalized with Dcx-staining since BrdU was injected 1 month before the analysis. Although the dose of BrdU used in the study was relatively high, we labeled only limited population of cells, which incorporated BrdU during their division and then started to differentiate long before the behavioral training. We can assume that all BrdU-labeled cells, which survived at the time of the start of behavioral experiment, ended their differentiation and, thus, stopped to express Dcx. Only cells differentiating during the period of the experiment end expressed Dcx; however, these cells appeared later than BrdU was available for labeling.

In the revised manuscript, Fig 2 shows data on PCNA staining and analysis, while Fig. 3 shows Dcx, BrdU+Dcx staining and respective analysis.

  1. In Figure 4, I or most readers would like to know the relationship between nNOS+ cells and NSPCs. Please provide high magnification confocal images demonstrating the identity of nNOS+

In the corrected MS, data on NOS expression are presented in Fig. 5. According to your commentary and in order to strengthen our opinion, we performed additional staining for nNOS and PCNA. These data are presented in Fig. 5 c and d. Similar to BrdU staining higher number of PCNA-positive cells was observed in 4-month-old rats (fig. 5c) as compared to 24-month-old rats (fig. 5d). It can be seen that nNOS-positive cells and their processes were located in close vicinities to the proliferating, i.e. PCNA-containing cells (the locations are indicated by asterisks in both Fig. 5c and d). We cannot say anything about the relationship between the nNOS-positive cells and stem cells because we did non study this in the present series of experiments.

  1. The authors hypothesized that the declined neurogenesis in aged hippocampus is related to increased local production of nitric oxide. Unfortunately, the authors did not test such an interesting hypothesis.

We fully agree with the Reviewer that to test a hypothesis on possible modification of local NO production is an interesting aim. Unfortunately, the only tool to test this is using nNOS knockout mice that was previously performed. In rats, only neuropharmacological approach based on the available inhibitors is possible. However, as we mentioned in the paper, this approach allowed to get quite controversial results. Most of the inhibitors interact with all NOS isoforms and have low specificity to nNOS, in particular after systemic administration. More specific nNOS inhibitors have low water solubility or are unavailable on the market.  This is mentioned in the chapter Limitations of the study.

Minor editing of English language required.

We tried to do our best to improve English. The MS was checked using “Grammarly: Free writing AI assistance” (www.grammarly.com)  

Reviewer 2 Report

In this manuscript, the age-related changes in the hippocampal neurogenesis in rats training in a water maze were assayed. The authors made an important conclusion that inhibition of neurogenesis may be associated with the increasing of local NO production. Overall, the data is competently presented and interesting enough, but I do have some suggestions that may enhance the presentation. I present these in no particular order of importance.

Fig. 4. Image artefacts on the top and right borders from both photos should be removed. Images from the hippocampal SGZ, CA4 and i.m.l. zones from 4 and 24-month-old rats should be added. In the figure legend, please indicate DAPI staining (It can be seen that the blue channel was also used). IR neurons (red and green) should be labeled (for example, with arrowheads).

Discussion. Please add some more recent references about the increasing of NO synthesis in the CNS with aging, for example:

Moiseev KY, Vishnyakova PA, Porseva VV, Masliukov AP, Spirichev AA, Emanuilov AI, Masliukov PM. Changes of nNOS expression in the tuberal hypothalamic nuclei during ageing. Nitric Oxide. 2020 Aug 1;100-101:1-6. doi: 10.1016/j.niox.2020.04.002.

Methods. A protocol for immunofluorescent staining should be described more precisely. Please, indicate the primary antibody and its dilution for BRDU staining. There are no data about visualization. Please, provide the type of microscope, filters for fluorescent microscopy and CCD camera.

Author Response

We thank the Reviewer for careful reading the manuscript and important notes made. Below, please find the response to the Reviewer’s criticism and information on the changes we included in the manuscript (highlighted in magenta).

Fig. 4. Image artefacts on the top and right borders from both photos should be removed. Images from the hippocampal SGZ, CA4 and i.m.l. zones from 4 and 24-month-old rats should be added. In the figure legend, please indicate DAPI staining (It can be seen that the blue channel was also used). IR neurons (red and green) should be labeled (for example, with arrowheads).

Fig. 4 was modified and now it is present as Fig. 5. The caption was corrected and DAPI staining was indicated.

Discussion. Please add some more recent references about the increasing of NO synthesis in the CNS with aging, for example:

Moiseev KY, Vishnyakova PA, Porseva VV, Masliukov AP, Spirichev AA, Emanuilov AI, Masliukov PM. Changes of nNOS expression in the tuberal hypothalamic nuclei during ageing. Nitric Oxide. 2020 Aug 1;100-101:1-6. doi: 10.1016/j.niox.2020.04.002.

We added several recent articles, including this one suggested by the Reviewer, on the relationship between NO synthesis and aging, including neurogenesis. 

Methods. A protocol for immunofluorescent staining should be described more precisely. Please, indicate the primary antibody and its dilution for BRDU staining. There are no data about visualization. Please, provide the type of microscope, filters for fluorescent microscopy and CCD camera.

These important details were added to the Materials and Methods chapter.

Round 2

Reviewer 1 Report

I do NOT find any significant improvements in the revised manuscript.

Minor editing of English language required

Reviewer 2 Report

The authors have improved their manuscript through the first round of revision by addressing all the points reviewers raised. I recommend accepting this manuscript for the publication.

Minor point: Please, delete the 1st line in the Abstract (line 10). It is from template.